# The Golgi Apparatus: A Voyage through Time, Structure, Function and Implication in Neurodegenerative Disorders

**DOI:** 10.3390/cells12151972

**Published:** 2023-07-31

**Authors:** Aurel George Mohan, Bogdan Calenic, Nicu Adrian Ghiurau, Roxana-Maria Duncea-Borca, Alexandra-Elena Constantinescu, Ileana Constantinescu

**Affiliations:** 1Department of Neurosurgery, Bihor County Emergency Clinical Hospital, 410167 Oradea, Romania; aurel.mohan@yahoo.com; 2Faculty of Medicine, Oradea University, 410610 Oradea, Romania; 3Immunology and Transplant Immunology, Carol Davila University of Medicine and Pharmacy, 020021 Bucharest, Romania; ileana.constantinescu@imunogenetica.ro; 4Centre of Immunogenetics and Virology, Fundeni Clinical Institute, 022328 Bucharest, Romania; 5Department of Surgical Disciplines, Faculty of Medicine and Pharmacy, University of Oradea, 410610 Oradea, Romania; ghiurau.adrian@yahoo.com; 6Forensic Medical Service, 140090 Alexandria, Romania; roxanaduncea@yahoo.com; 7Faculty of Medicine, Carol Davila University of Medicine and Pharmacy, 020021 Bucharest, Romania; alexandra-elena.constantinescu0720@stud.umfcd.ro

**Keywords:** Golgi apparatus, neurodegenerative diseases, Parkinson’s disease, endoplasmic reticulum stress, neurobiology, Golgi disfunction

## Abstract

This comprehensive review article dives deep into the Golgi apparatus, an essential organelle in cellular biology. Beginning with its discovery during the 19th century until today’s recognition as an important contributor to cell function. We explore its unique organization and structure as well as its roles in protein processing, sorting, and lipid biogenesis, which play key roles in maintaining homeostasis in cellular biology. This article further explores Golgi biogenesis, exploring its intricate processes and dynamics that contribute to its formation and function. One key focus is its role in neurodegenerative diseases like Parkinson’s, where changes to the structure or function of the Golgi apparatus may lead to their onset or progression, emphasizing its key importance in neuronal health. At the same time, we examine the intriguing relationship between Golgi stress and endoplasmic reticulum (ER) stress, providing insights into their interplay as two major cellular stress response pathways. Such interdependence provides a greater understanding of cellular reactions to protein misfolding and accumulation, hallmark features of many neurodegenerative diseases. In summary, this review offers an exhaustive examination of the Golgi apparatus, from its historical background to its role in health and disease. Additionally, this examination emphasizes the necessity of further research in this field in order to develop targeted therapeutic approaches for Golgi dysfunction-associated conditions. Furthermore, its exploration is an example of scientific progress while simultaneously offering hope for developing innovative treatments for neurodegenerative disorders.

## 1. Brief Historical Background

Organelles, which are intracellular membrane-bound structures found within eukaryotic cells, have been described since light microscopy was invented and cell theory was established during the 19th century. Subcellular fractionation allowed researchers to further discover new organelles, while radiolabeling techniques allowed modern experiments on organelle biogenesis to begin taking place simultaneously.

Camillo Golgi created his “black reaction”, or Golgi stain, while serving as chief physician of a hospital near Milan in 1873. This method allowed for an accurate overview of single nerve cells [1]. As part of his research, Golgi noted a unique reticular structure within the cytoplasm of nerve cell bodies distinct from their cell membrane and nuclei; however, this structure could not be consistently stained. Emilio Veratti successfully replicated and confirmed these findings in the cell bodies of the fourth cranial nerve in Golgi’s laboratory, prompting Golgi to continue with neurocytological research, eventually achieving excellent and reproducible staining of Purkinje cells of the Tyto alba species of owl. At a meeting of the Medical-Surgical Society of Pavia in April 1898, Golgi confidently described a “fine and elegant network” internal to nerve cells. He stressed ribbon-like threads’ manner of division and connection (anastomoses) and the pathways they formed. Golgi also observed thin plates or small transparent discs at nodal points of the network as its most notable characteristic with clearly defined outer surfaces yet penetrating deeper planes within the cell body interior [2,3,4].Golgi coined this structure “apparato reticolare interno” due to its shape and intracellular location. While Golgi initially identified this apparatus in Purkinje cells, he believed that its existence may be shared among major classes of nerve cells. Emilio Veratti established the existence of the cytoplasmic network known as the sarcoplasmic reticulum in 1902 by showing its presence in the 4th cranial nerve. On 19 April 1898, Veratti presented this structure formally for the first time at the Pavia Medical-Surgical Society as an “internal reticulated structure,” comprising ribbon-like filamentous elements that anastomose, small plates with clear centers serving as nodal points, and rounded discs arranged in various combinations. Antonio Pensa, working at Camillo Golgi’s General Pathology and Histology Laboratory in 1899, first identifiedthis organelle in adrenal medulla cells. Shortly afterwards, Adelchi Negri (then a medical student) identified similar structures in the thyroid, epididymis, salivary glands, ovarian cells, and nerve cells, later discovering intraneuronal inclusions known as “Negri bodies” while studying rabies-infected brains. Edoardo Gemelli, one of Golgi’s students, discovered a similar structure in pituitary gland cells and hypothesized that this network might play an integral part of secretory or nutritional processes within cells. Camillo Golgi changed the staining technique originally defined by Ramon y Cajal and developed his own in 1903. By using this new approach, Golgi could observe both the morphology and localization of the Golgi apparatus during its secretion process in the mucous glands within the stomach of frogs. Over the late 19th and early 20th centuries, scientists investigated the localization and physiological functions of Golgi apparatus (GA) [5,6]. Camillo Golgi, after whom it is named, observed that GA organelles were located apically above nuclei. At approximately the same time, Giuseppe D’Agata also investigated reticular networks within gastric epithelia. Prior to Golgi’s work between 1867 and 1887, several scientists made mention of oreticular structures within cells, either before or after his introduction of this organelle. It is possible that some of these researchers were also keeping an eye on the GA [7,8,9]. However, for many years the GA was closely associated with the University of Pavia, where Golgi and his students published over 70 articles about it, detailing its changes under different developmental, physiological, and pathological conditions as well as its presence across cell types. Therefore, GA was found to be highly variable and unstable after extensive studies; even so, its authenticity came under scrutiny by subsequent researchers, some of whom saw it as the product of fixation techniques or metallic impregnation techniques. George Palade and Albert Claude of the Rockefeller Institute later observed similar cytoplasmic structures in various cells without using specific staining methods or applying 40–55% ethanol, suggesting that GA represented one or more artificially induced myelin figures during sample preparation. Such debates continued even when GA could not be detected via electron microscopy examinations [10].

The Golgi apparatus (GA) was considered unreliable until electron microscopy came along in the mid-1950s and provided new insight. Dalton, Felix, and others published electron micrographs of the Golgi complex, revealing its cisternae and vesicles that stain similarly under light microscopy. Electron microscopy revealed a large variety of vesicles within cells, representing components of secretory and endocytic pathways. Jamieson and Palade provided us with our first insight into the dynamic nature of vesicles and organelles when they employed radioactive tracers and electron microscopic autoradiography in 1967. Their research showed that newly synthesized secretory proteins traveled from the rough endoplasmic reticulum through the Golgi region and eventually into secretory granules [11]. By the 1980s, it had become widely accepted that endocytosed vesicles fuse with an intracellular organelle known as an endosome for recycling back to plasma membrane recycling sites, delivery to lysosomes, or transportation to trans-Golgi networks. Vesicular trafficking has emerged as the central mechanism for protein transport along the secretory and endocytic pathways [12,13]. In the 1980s, significant progress was made toward understanding the molecular machinery underlying vesicular traffic, including reconstituting individual steps from animal cells in cell-free systems and discovering secretory mutants in yeast. Reconstitution of Golgi traffic between Golgi cisternae was particularly instructive: incubating Golgi membranes with cytosol and ATP set the scene for further examination into the molecular mechanisms underlying vesicular traffic. These developments marked an entryway into studying the molecular mechanisms underlying this form of transport [14].

An important experimental system involved co-incubating Golgi membranes from cells lacking N-acetylglucosamine transferase but containing VSV G protein with wild-type Golgi membranes to induce vesicular traffic and radioactive labeling of VSV-G with N-acetylglucosamine via transferase activity. This assay revealed COPI (coat protein I)-coated vesicles as well as components of general cytosolic fusion machinery necessary for both secretory and endocytic pathways [15].

Subsequently, it became evident that cells possess two classes of clathrin-coated vesicles: one located predominantly within the Golgi for budding from the trans-Golgi network, and another situated at the plasma membrane and responsible for major pathways of endocytic uptake. These two categories of clathrin-coated vesicles could be differentiated by the presence of two heterotetrameric adaptor protein complexes: AP-1 in the trans-Golgi network and AP-2 at the plasma membrane. Over time, more coat proteins were discovered, including heterotetrameric AP-3 and AP-4 complexes that do not associate with clathrin and distantly related monomeric GGAs (Golgi-localized, g-ear-containing ARF binding proteins). All these coat proteins play roles in post-Golgi membrane traffic pathways [16,17].

## 2. Organization and Structure

The Golgi apparatus, an organelle commonly found in plant, animal, and fungal cells, consists of membrane-bound organelles. One unique characteristic of its organization is the flat, perforated cisternae that organize these organelles. Most single-celled lower eukaryotes typically possess only a single Golgi stack per cell, while plants and Drosophila contain multiple scattered Golgi stacks within their cytosol, and mammalian cells usually feature several interconnected stacks connected together by narrow channels that form ribbon-like structures near their nuclei [18]. Though stacking of the Golgi apparatus may not be strictly necessary for its function in budding yeast, its widespread presence across multicellular and unicellular eukaryotes suggests significant functional implications. The Golgi complex plays a crucial role in protein processing, sorting, and transport within cells, acting like a factory where proteins from the endoplasmic reticulum (ER) undergo further modification before being sent outward to their destinations, such as lysosomes, plasma membrane or secretion sites [19,20].

The Golgi apparatus plays a critical role in synthesizing glycolipids and sphingomyelin for secretion, as well as other important substances like neurotransmitters. Functionally, its four distinct regions include the cis-Golgi network; the Golgi stack (comprising medial and trans subcompartments); the trans-Golgi network; and the Golgi organ. Proteins transported from the ER enter the Golgi apparatus through an intermediate compartment called the ER-Golgi intermediate compartment and enter at the cis-Golgi network before continuing through the medial and trans compartments of Golgi stack, where most metabolic activities take place. Finally, modified proteins, lipids, and polysaccharides move on to the trans-Golgi network, which acts as a central sorting and distribution center, channeling molecular traffic towards either lysosomes, the plasma membrane, or extracellular spaces [21]. Golgi apparatus was one of the first organelles observed and described by early light microscopists due to its highly visible structure resembling that of pancakes. Golgi stacks consist of four to six cisternae; however, certain unicellular flagellates may contain up to 60. Animal cells often interconnect multiple Golgi stacks through tubular connections that link corresponding cisternae into one complex structure. Golgi apparatus are often found close to both the cell nucleus and centrosome in animal cells, although in animal cells their positioning relies on microtubules. Experimental depolymerization of microtubules causes Golgi apparatus reorganization to occur, with individual stacks dispersing throughout the cytoplasm near sites of ER exit. Conversely, many cells, such as plant cells, contain many scattered Golgi stacks naturally within their cytoplasms [22,23].

As molecules pass through the Golgi apparatus, they undergo an intricate sequence of covalent modifications. Each Golgi stack has two distinct faces: the cis face (entrance side) and the trans face (exit side). Substantially attached to these surfaces are special compartments known as cis-Golgi networks (CGN) and trans-Golgi networks (TGN). Both networks consist of interlinked tubular structures with cisternal chambers; their primary role is protein sorting [24]. Proteins and lipids move from their origins in the endoplasmic reticulum (ER) through tubular clusters in the CGN to exit via the TGN to ultimately reach the cell surface or other compartments for final sorting processes. Both organelles play an integral part in this sorting process. Proteins entering the CGN can either progress towards the Golgi apparatus or be returned retrogradely to the ER for recycling. Proteins released from the TGN may either continue along their designated paths toward lysosomes, secretory vesicles, or cell surface organelles; or be retrogradely transported backwards. Goblet cells of the intestinal epithelium produce large quantities of polysaccharide-rich mucus that enters the digestive tract via goblet cells of the Golgi apparatus and secretes significant quantities into its lumen. Notably, these cells feature large vesicles on their trans side, corresponding to where secretion takes place in the plasma membrane regions of secretion [25,26,27,28,29,30].

## 3. Functions and Main Roles

### 3.1. General Organization of the Golgi Complex

The Golgi apparatus (sometimes referred to as Golgi complex) plays an essential role in cell biology. It serves as a primary site for carbohydrate synthesis as well as an intermediate storage site for products originating from the endoplasmic reticulum (ER). The Golgi apparatus plays an essential role in synthesizing carbohydrates such as plant cell wall components such as pectin and hemicellulose as well as extracellular matrix glycosaminoglycans for animals, all within its confines [30]. The Golgi apparatus acts as a hub for processing and modifying proteins and lipids received from the ER, acting simultaneously. Many molecules found in the Golgi are decorated with oligosaccharide side chains to aid their transport in vesicles bound for lysosomes. Specific oligosaccharide groups serve as tags to direct these proteins towards specific transport vesicles bound for Lysosomes. However, for most proteins and lipids, once they obtain their appropriate oligosaccharides from the Golgi apparatus, they are recognized by other mechanisms and targeted into transport vesicles that will deliver them directly to their desired destinations within cells [21,31].

### 3.2. Protein Processing and Transport

The Golgi apparatus plays an essential role in the modification and synthesis of carbohydrates within glycoproteins. One aspect of this process involves altering N-linked oligo saccharides initially added to proteins by the endoplasmic reticulum (ER), such as adding 14 sugar residue oligosaccharides; additionally, three glucose residue and one mannose residues may be removed prior to transportation to the Golgi apparatus, where their N-linked oligosaccharides undergo extensive modification through sequential reactions [32].

For proteins destined for secretion or plasma membrane release, Golgi modification usually begins by stripping away three additional mannose residues and adding an N-acetylglucosamine, then gradually stripping away two more mannose residues and adding fucose and two N-acetylglucosamines; finally, three galactose and three sialic acid residues are added. The amount of modification can differ for various glycoproteins depending on factors like their structure as well as processing enzyme presence within Golgi complexes belonging to different cell types, with proteins emerging with various N- linked oligosaccharides attached to each exiting the Golgi complex, resulting in proteins exiting with various N-linked oligosaccharides linked by each cell type; therefore, proteins exit the Golgi with various N-linked oligosaccharides attached [33,34].

Apart from its role in processing and sorting glycoproteins, the Golgi apparatus also plays a significant role in lipid metabolism by producing glycolipids and sphingomyelin synthesis [35]. As previously noted, the ER synthesizes glycerol phospholipids, cholesterol, and ceramide. Following that step, however, the Golgi apparatus plays its part by producing nonglycerol phospholipids from these precursors, such as sphingomyelin, or by adding carbohydrates to these precursors to produce various kinds of glycolipids that exist within cell membranes [35]. Sphingomyelin production from nonglycerol phospholipids found only within cell membranes by adding carbohydrates to ceramide can generate various types of glycolipids from this precursor compound [36]. Proteins, lipids, and polysaccharides are transported from the Golgi apparatus to their final destinations via the secretory pathway. This process entails sorting proteins into various transport vesicles that bud from the trans-Golgi network and carry their cargo directly to specific cellular locations [37]. Some proteins are transported directly from the Golgi apparatus to the plasma membrane via the constitutive secretory pathway, which integrates new proteins and lipids into the plasma membrane and continuously releases proteins out of cells. To initiate the biosynthetic-secretory pathway, proteins entering the endoplasmic reticulum (ER) that are headed toward the Golgi apparatus or further destinations are packaged into small transport vesicles coated with COPII proteins for transport [37]. These vesicles bud off from specific regions called “ER exit sites”, which lack ribosomes on their membranes and, in most animal cells, can be found randomly distributed throughout their network [38].

Initial assumptions suggested that all proteins not attached to the ER would automatically enter transport vesicles for transport; however, recent research has shown that packaging into transport vesicles leaving the ER can also be a selective process [39,40]. Certain cargo proteins are actively recruited into these vesicles and become concentrated. It has been suggested that these cargo proteins contain specific exit signals on their surfaces that are recognized by complementary receptor proteins. These receptor proteins become trapped within budding vesicles by interacting with components of the COPII coat. Proteins without exit signals may still be packaged into vesicles and leak slowly from ER-resident protein pools over time.

Furthermore, secretory proteins with high abundance may leave the ER without needing sorting receptors to exit; the nature of their exit signals that take them towards Golgi transport remains largely unknown, with one notable exception. ERGIC53 acts as a receptor to package certain secretory proteins into COPII-coated vesicles for transport. Its role was highlighted when individuals carrying an inherited mutation depriving them of ERGIC53 showed reduced levels of two blood-clotting factors (Factor V and Factor VIII), leading to excessive bleeding [41,42].

The ERGIC53 protein acts as a mannose-binding lectin that recognizes and binds to mannose residues present on Factor V and Factor VIII proteins, helping facilitate their packaging into transport vesicles within the ER [42]. For proteins to leave the ER, they must undergo proper folding. When part of multimeric complexes, full assembly may also be required. Misfolded or incompletely assembled proteins remain within the ER, where they interact with chaperone proteins such as BiP or calnexin [43]. These chaperones may mask exit signals on these proteins or otherwise anchor them within their location in the ER.

These defective proteins are then transported back into the cytosol, where they are degraded by proteasomes. Quality-control mechanisms are of utmost importance, as misfolded or improperly assembled proteins could interfere with other proteins’ normal function transported further along the secretory pathway, necessitating extensive corrective actions [43]. Over 90% of newly synthesized subunits of proteins such as the T cell receptor and acetylcholine receptor tend to be degraded within cells before reaching their functional roles on cell surfaces, necessitating cells to produce an abundance of different protein molecules so as to select those that assemble correctly and thus fulfill their functions [44,45]. Once transport vesicles have undergone coat shedding after budding from an ER exit site, they become vulnerable to membrane fusion with other transport vesicles in their same compartment. This event, known as homotypic fusion, differentiates it from heterotypic fusion, which involves joining membranes from different compartments together [46]. Both events rely on specific sets of SNARE proteins matching up perfectly between membranes; however, both membranes contribute their own sets of v-SNAREs and t-SNAREs (Figure 1) [47].

Vesicular tubular clusters are created when vesicles from the ER fuse with one another. When observed under an electron microscope, they appear as convoluted structures. Although distinct from ER cells in that they lack several proteins typically present therein, continuously generated clusters serve as transport packages responsible for carrying materials from the ER to the Golgi apparatus; their lifespan, however, tends to be fairly short as they quickly move along microtubules towards the Golgi apparatus, where they leave their content [48].

In the forming process of tubular clusters, there is an interesting moment where these clusters give rise to budding vesicles of their own; unlike COPII-coated ER vesicles that bud from it, these vesicles coated with COPI proteins serve the purpose of retrieving resident proteins that have escaped from ER, as well as proteins involved in the initial budding process that require returning. This retrieval mechanism highlights precise control mechanisms governing coat assembly reactions, specifically why assembly of COPI coat begins moments after COPII coat shedding occurs; its exact regulatory mechanism remains unknown. Retrieval transport, commonly referred to as retrograde transport, occurs as the vesicular tubular clusters move towards the Golgi apparatus and gradually change in composition as specific proteins are retrieved and sent back towards the ER. A similar process also takes place following delivery by the tubular clusters themselves [39,49,50].

### 3.3. Lipid Metabolism

Sphingomyelin production takes place on the inner side of Golgi membranes, while glucose is added to ceramide on its cytosolic side. However, some reports indicate that glucosylceramide undergoes flipping, with additional carbohydrates being added on the luminal side of the membrane. Sphingomyelin and glycolipids cannot cross the Golgi membrane, so they remain localized within its luminal half. Once transported by vesicles, glycolipids become localized on the outer leaflet of the plasma membrane, with their polar head groups exposed on cell surface. Oligosaccharide components in glycolipids serve as important surface markers in cell recognition [51]. Plant cells rely heavily on their Golgi apparatus as the site for producing the polysaccharides present in their cell wall. Cellulose, one of three major polysaccharide constituents found within plant cell walls, is produced at its surface via enzymes found on the plasma membrane [52].

On the other hand, polysaccharides found in cell walls, such as hemicelluloses and pectins, are more complex molecules with branching structures, synthesized in the Golgi apparatus before being transported out through vesicles to reach their destinations on cell surfaces. Their production constitutes an essential cellular process; up to 80% of plant cell Golgi metabolism activity goes towards their synthesis [53].

### 3.4. Polysaccharide Metabolism

The Golgi apparatus plays an essential role in the modification and synthesis of carbohydrate portions of glycoproteins, with one primary function being the modification and syn thesis of their carbohydrate portions. A key aspect of this process involves the modification of N-linked oligosaccharides that were initially attached to proteins by the ER; specifically, an oligosaccharide composed of 14 sugar residues is added before three glucose residues and one mannose residue are removed before being transported for processing into the Golgi apparatus, where their N-linked oligosaccharide undergo extensive further modifications [54]. The Golgi apparatus processes N-linked oligosaccharides through a sequence of reactions. One initial modification involves stripping off three additional mannose residues from proteins bound for secretion or plasma membrane transport. After this step is completed, N-acetylglucosamine, two mannoses, and fucose are removed, followed by the addition of three galactose and sialic acid residues. Noteworthy is the fact that different glycoproteins undergo differing degrees of modification as they pass through the Golgi complexes of different cell types, depending on both their structure and the presence or absence of processing enzymes present therein. As such, proteins can emerge with various N-linked oligosaccharides attached at their ends after exiting [55].

Lysosomal proteins take a different approach when processing N-linked oligosac-charides than secreted and plasma membrane proteins; rather than initially removing three mannose residues as part of their modification process, lysosomal proteins undergo mannose phosphorylation instead. The starting point of this process involves attaching N-acetylglucosamine phosphates to specific mannose residues located within the cis-Golgi network. Subsequently, the N-acetylglucosamine group is removed, leading to mannose-6-phosphate residues on the N-linked oligosaccharide that are not removed during further processing like with other proteins. Mannose-6-phosphate receptors located in the trans-Golgi network play an essential role in providing specific recognition of modified proteins for transport to lysosomes, where they will eventually become functional components and fulfill their roles [56].

O-linked glycosylation refers to modifying proteins by attaching carbohydrates directly to specific serine or threonine residues of certain amino acid sequences (referred to as O-linked glycosylation). This process occurs within the Golgi apparatus by the sequential addition of sugar residues; typically, serine/threonine are directly linked with N-acetylga-lactosamine for direct linking before other sugars may be added as required, including even further modifications such as adding sulfate groups if required [57,58].

#### Position of Enzymes Engaged in Glycosylation within Golgi Subdivisions

Glycosylation processes in the secretory pathway follow an organized and sequential progression, often involving glycosyltransferase reactions. These enzymes, along with their glycan substrates attached to proteins or lipids, and the appropriate nucleotide sugar donor, must reside in the same compartment. Biochemical and ultrastructural analyses have revealed that glycosyltransferases cluster into distinct yet interlocking compartments along the secretory pathway. Enzymes involved in early biosynthetic pathways tend to reside in the cis- and medial-Golgi compartments, while those involved in later stages usually reside in trans-Golgi cisternae or the trans-Golgi network (TGN). These observations have resulted in extensive investigation of how glycosyltransferases and processing glycosidases can achieve compartmental segregation of their activities. Early studies focused on identifying enzyme sequences responsible for their retention within Golgi cisternae using the vesicular transport model of protein trafficking. More recent investigations have incorporated into their framework the Cisternal Maturation Model in order to gain a broader understanding of these mechanisms [59,60].

Understanding how proteins move through the Golgi stack and its enzymes maintain their positions within its cisternae has greatly advanced in recent years. Two primary models—neither mutually exclusive—have been proposed that may function together. The Vesicular Transport Model suggests that the Golgi apparatus serves as stable compartments in which cargo proteins are transported via coated vesicles from the endoplasmic reticulum (ER) to an intermediate compartment and between the Golgi cisterna in an orderly fashion. Golgi glycosylation enzymes play an essential role in this process, altering cargo proteins at every cisterna of the Golgi. A recent discovery supports the cisternal maturation model, which describes the transport of large cargo molecules through the Golgi that cannot fit in small transport vesicles [61].

Studies demonstrate the significance of the cisternal maturation model on Golgi enzyme localization by investigating the role of Conserved Oligomeric Golgi (COG) complex proteins in retrograde transport of cargoes. Results obtained indicate that mutations to COG subunits have an impactful influence on Golgi enzyme distribution as well as overall protein glycosylation [62]

The COG complex, consisting of eight subunits, serves as a cytoplasmic tethering complex that connects incoming vesicles with their target compartments for fusion. Working alongside COPI subunits, it also facilitates retrograde transport within Golgi cells as well as from Golgi to the endoplasmic reticulum (ER) [63].

Mutations that affect COG subunits may result in Golgi glycosyltransferases that act improperly, leading to anomalies in glycosylation processes. While not directly interacting with Golgi enzymes, the COG complex plays an essential role in retrograde vesicular transport within Golgi systems for efficient glycosylation. Mutations in COG subunits have been linked with Congenital Glycosylation Disorder type II (CDG-II) [64].

Studies of mutant and chimeric Golgi enzymes have demonstrated that different enzymes require distinct conditions for their localization. Initial research focused on TM regions of enzymes like GlcNAcT-1 (located in the medial Golgi) [65], GalT-1 (found in the trans Golgi) [66], and ST6Gal-I (found both trans Golgi network and the medial-Golgi) [67]. Subsequent investigations have demonstrated that many signals and mechanisms contribute to the localization of enzymes. Homo- and hetero-oligomerization play an integral part in localizing certain Golgi enzymes; additionally, there is strong evidence for their retrograde transport into Golgi organelles via glycosyltransferase cytoplasmic tails. Transmembrane (TM) regions play an essential role in partitioning membrane microdomains and ultimately trafficking and localizing proteins across cells. Their length and hydrophobicity determine this. Cholesterol concentration and membrane width increase along the secretory pathway, with the widest and most cholesterol-rich membranes, found at cell surfaces [68]. Studies conducted using cholesterol-containing model membranes have demonstrated that shorter TM peptides tend to partition into thinner membranes while thicker membranes favor longer TM peptides. This finding suggests that cholesterol, with its ability to straighten acyl chains of lipids, might energetically prevent the partitioning of TM peptides into membranes with mismatched thicknesses. Furthermore, these findings support the notion that membrane thickness could influence the localization of membrane proteins within the secretory pathway [69].

Evidence supporting the relationship between membrane thickness and protein localization includes the observation that endoplasmic reticulum (ER) proteins typically possess shorter transmembrane domain regions compared with plasma membrane proteins. Golgi enzyme TM regions fall between those found on ER and plasma membrane proteins in terms of length; it should be noted, however, that Golgi enzymes do not display an increase in length from their cis to trans organelle surfaces. The impact of TM region length on cisternal localization could differ depending on other sequences and mechanisms involved with enzyme localization. Still, Golgi enzymes with shorter TM regions likely remain within the Golgi system by restricting their ability to become part of thicker, cholesterol-rich membrane transport carriers destined for post-Golgi compartments such as the plasma membrane. The location of enzymes within the Golgi is also determined by their ability to form multimeric complexes [69]. Most enzymes involved with N-linked and O-linked glycosylation pathways form homodimers; many can even form het eromeric complexes. The formation of heteromeric complexes may be pH-dependent. Such formation occurs between enzymes catalyzing sequential reactions within one pathway and located within close proximity in separate cisternae. On the N-glycosylation pathway, complexes are created between N-acetylglucosaminyltransferases GlcNAcT-I and Glc-NAcT-II in the medial-Golgi region and GalT-I and ST6Gal-I in the trans-Golgi region [66,67]. However, enzymes from different pathways (O-glycosylation and N-glycosylation enzymes) or within one pathway that compete or have nonsequential events do not form heteromeric complexes, even if localized in the same cisterna. The formation of complexes between sequential enzymes may increase efficiency by facilitating substrate channeling, where one enzyme transfers newly modified substrate directly to the next enzyme in the pathway. As evidence suggests, glycosylation enzymes utilize multiple mechanisms to remain localized within the Golgi apparatus. A combination of signals and mechanisms utilized by an enzyme may determine its stability within its Golgi host compartment, its potential transition into later compartments, or whether or not cleavage and secretion into extracellular spaces occur [54,70,71].

## 4. Golgi Apparatus–Biogenesis

Golgi complexes, an essential organelle of cells, are passed down from parent cell to daughter cell through duplication of existing stacks, with each daughter receiving its own Golgi stack. Two main mechanisms for creating new Golgi structures have become standard practice: de novo formation and templated growth [72].

The de novo formation model postulates that new Golgi structures arise independently of existing ones, using materials from the endoplasmic reticulum (ER). Existing Golgi structures do not play any direct part in this process. On the other hand, the templated growth model proposes that new Golgi structures arise through lateral expansion followed by medial fission from existing ones through an indirect process known as templated growth. Here, existing Golgi apparatuses serve as templates for the growth of new Golgi apparatus structures; both sources contribute equally to producing them [73].

Organisms such as Toxoplasma gondii and Trichomonas foetus use templated growth to form new Golgi structures, using existing Golgi structures as both templates and materials for this formation. Lateral growth may be enhanced by the accumulation of lipids and proteins from their environment or produced locally within the Golgi apparatus itself, with accumulation being further promoted through medial fission during cell division, although this remains poorly understood [74,75].

Pichia pastoris stands out among fission yeasts with its distinctive Golgi formation properties. Unlike Saccharomyces cerevisiae, which lacks stackable Golgi compartments per cell, Pichia pastoris contains both compartments, which form spontaneously at their respective exit sites from the ER and may spawn stackable Golgi apparatus compartments by de novo formation or templated growth to ensure adequate Golgi localization and function during cell division. These mechanisms illustrate different strategies cells use to ensure proper Golgi localization and function during cell division [59,76,77].

Time-lapse video microscopy and GFP-tagged proteins during mitosis have revealed that Golgi stacks often fuse during mitosis, yet their average size remains constant due to shrinkage post-fusion and de novo formation. Furthermore, as cells divide, they possess at least two Golgi stacks before division occurs [78].

Trypanosoma brucei cells that possess one Golgi stack located adjacent to their basal body utilize both de novo biogenesis and templated growth mechanisms for Golgi assembly _474_: De novo synthesis relies heavily on material delivered from the ER exit site; cell division necessitates duplication; proteins from the Centrin family, such as Centrin1 and Centrin2, are thought to assist this duplication process, while Centrin2-containing structures may mark sites where new Golgi assembly takes place; their exact composition remains elusive [79]. Trypanosoma has an unusual hybrid model in which its new Golgi apparatus formation involves both new material and transfers from existing Golgi. This suggests a hybrid form of biogenesis: old Golgi materials may act as seeds or templates for de novo assembly while an ER exit site supplies essential lipids and proteins needed for its creation. Further investigations will need to be conducted into each element’s impact on Golgi inheritance across different model systems [80,81,82].

## 5. Golgi Apparatus in Neurological Diseases

The Golgi apparatus is an organelle within cells that adapts and changes its structure based on the physiological status of cells. One prominent structural change is Golgi fragmentation, in which stacks disperse or completely disassemble; this phenomenon may even precede cell apoptosis, as seen by an in vitro model of mechanical cell injury; it often precedes rather than follows cell death.

Pharmacological or oxidative stress can trigger various changes to the Golgi apparatus, including cargo overload, ionic imbalance, disruptions in glycosylation, luminal pH changes, and altered glycosylation processes. This results in inadequate Golgi glycosylation and impaired membrane trafficking, resulting in Golgi stress. When cells perceive this state, they activate autoregulatory mechanisms designed to repair their Golgi apparatus while initiating signaling pathways designed to mitigate any associated anxiety. The procaspase-2/golgin-160 pathway involves procaspase-2 interacting with upstream apoptotic regulators to cleave Golgi matrix proteins like golgin-160 and giantin, both with nuclear localization signals, which then enter the nucleus and participate in the stress repair pathway [83,84]. Another pathway, TFE3, involves dephosphorylating and nuclear translocating TFE3, leading to transcription of genes containing Golgi apparatus stress elements and increasing expression of glycosylation enzymes as well as components involved in membrane trafficking. HSP47 plays an essential role in Golgi stress; when downregulated, HSP47 causes Golgi fragmentation that leads to stress-induced apoptosis. Meanwhile, the CREB3/ARF4 pathway regulates ARF4, which localizes to Golgi membranes and controls traffic between the Golgi and endoplasmic reticulum (ER) [85,86,87]. Under Golgi stress conditions, low levels of ARF4 lead to nuclear translocation and activation of CREB3, followed by increased transcription of ARF4 [88]. If these repair pathways fail, however, complete _509_ disassembly of the Golgi apparatus ensues, leading to cell apoptosis. Structural modifications of the Golgi apparatus have been observed in various neurodegenerative diseases, including amyotrophic lateral sclerosis, Alzheimer’s disease, Parkinson’s disease, Huntington’s disease, Creutzfeldt-Jacob disease, and multiple system atrophy. Golgi fragmentation, often thought of as a late event of cell apoptosis, has recently been postulated as possibly occurring prior to microtubule degeneration and thus could act as an initiator of neuronal loss [89,90].

At first, neurodegenerative diseases typically trigger an adaptive response from the Golgi apparatus, such as compensatory expansion. Over time, however, when exposed to excessive excitotoxins and oxidative insults, its adaptive properties begin dissipating until atrophy sets in and it eventually disperses. Multiple mechanisms contribute to neuronal Golgi fragmentation during disease pathogenesis: (1) mutant growth hormones disrupt trafficking between endoplasmic reticulum (ER) and Golgi; and thirdly, the interruption of the ER-lysosome-Golgi network that regulates protein glycosylation leads to Golgi fragmentation; (2) impaired retrieval of membrane proteins from endosomes into Golgi [91,92]. Golgi glycosylation is essential to proper protein folding, increasing the hydrophilicity of intermediates and potentially preventing mutant protein aggregates, an integral aspect of neurodegeneration. Studies have demonstrated that inhibiting neuronal Golgi fragmenta tion reduces or delays cell apoptosis, suggesting that a fragmented Golgi apparatus in neurons significantly contributes to cell death and neurodegeneration. Oxidative stress signals, initially induce Golgi stress in response to oxidative stress signals which then leads to excessive stimulation that causes fragmentation of the Golgi apparatus, leading to amplified stress signaling and accelerating neuronal dysfunction resulting in neurodegeneration.

Neurogenesis occurs in two specific regions in adult mammalian brains known as neurogenic niches: the subventricular zone (SVZ) and the subgranular zone (SGZ). The latter, located within the dentate gyrus of the hippocampus, continuously generates new granule cells in adult individuals; these new neurons play key roles in memory formation and mood regulation. At each stage of granule cell development, one primary dendrite with branches reaching toward the molecular layer and one long axon projecting through the hilar region to the CA3 region form. Basal dendrites do not form in mature adult-born granule cells due to basal dendrite absence, thus providing vital clues as to the proper formation and integration of neural circuits and signals. Emerging evidence points towards the Golgi apparatus being involved in the establishment of polarized dendrites among adult-born granule cells. Studies conducted on Golgi-related genes within these neurons have identified STK25 and STRAD as regulatory proteins for neuronal development; disruption through knockdown may result in abnormal dendrite formation. Eight-week-old mice exhibit Golgi apparatus distribution primarily within the initial segment of a primary dendrite. Interference with its deployment by manipulating Golgi-related proteins, such as downregulating Golgi matrix protein GM130 or overexpressing GRASP65, can disrupt its deployment, potentially altering dendrite formation and extension in adult-born granule cells.

The Golgi apparatus plays an essential role in dendrite polarity formation among adult-born hippocampal granule cells, with its polarized distribution essential to the proper formation of dendrites. Furthermore, this organ is associated with neural stem and progenitor cells within other neurogenic regions like the subventricular zone (SVZ) and the ventricular zone as a way of modulating their identity and cytoarchitecture [93,94,95,96,97]. The polarized architecture of neural stem cells is inextricably tied to their Golgi apparatus’ asymmetric localization. Bipolar epithelial neural stem cells exhibit noncanonical features of their Golgi apparatus, such as Reelin, liver kinase B1 (LKB1), PITPNA/PITPNB, STK25, and Brefeldin A-inhibited guanine exchange factor 2 (BIG2) (Figure 2) [98,99,100,101,102]. A polarized distribution of the Golgi apparatus could contribute to dendrite polarity through post-Golgi membrane trafficking. Golgi-mediated membrane trafficking pathways facilitate protein movement from the cell body to dendrites, with disruptions to ER-to-Golgi trafficking impairing dendritic growth. Protein Kinases D1 and 2 (PKD1) play important roles in post-Golgi trafficking by controlling the directionality of Golgi derived vesicles. Impaired function of PKD1 and PKD2 in hippocampal neurons leads to mislocalized, specific vesicles, leading to the transformation of preexisting dendrites into axons. Under normal circumstances, post-Golgi membrane trafficking preferentially targets one principal dendrite instead, suggesting an association between anasymmetric Golgi apparatus distribution and dendritic outgrowth [100,103,104,105,106].

Therefore, changes to Golgi structures or localization during neurogenesis may have detrimental repercussions for their structural and functional development.

## 6. Golgi Apparatus–Parkinson

Parkinson’s disease (PD) is the second-most prevalent neurodegenerative condition worldwide, and its incidence is projected to rise over time. Characterized by both motor and non-motor impairments caused by dopamine deficiency, its pathogenesis remains complex, but environmental and genetic factors may play a part in its pathogenesis, as environmental and genetic influences may contribute to mitochondrial dysfunction, protein aggregation, oxidative stress exposure, autophagy dysfunction impairment, neuroinflam mation as well as mitochondrial dysfunction and dopamine deficiency [107]. Fragmentation of the Golgi complex (GC) is an early telltale sign of neurodegenerative diseases like Parkinson’s Disease (PD). Early postmortem analysis of PD samples demonstrated high levels of fragmentation. Alpha-synuclein (a-syn), an important protein implicated in its pathogenesis, aggregates intracellularly when damaged Golgi apparatuses lead to fragmentation; this causes inclusions to form that accumulate toxins, which lead to increased oxidative stress and cell death [108].

Apoptosis, in which both the structure and function of GAs are disrupted, plays an essential role in CNS pathological conditions. Studies of early-stage apoptotic cell death suggest GC fragmentation may not simply be the result of cell death due to apoptosis; however, its exact mechanism and role within this process remain enigmatic.

Leucine-rich repeat kinase 2 (LRRK2), Rabs, VPS52, and Cyclin-dependent Kinase 5 (CDK5) [109] have all been linked with Parkinson’s disease (as it can be seen in Figure 3). Of these proteins, LRRK2 stands out as being particularly prevalent, being one of the more frequently mutated genes associated with the condition-LRRK2 contains both GTPase domains; mutations are present both with autosomal dominant Parkinsonism as well as being linked with its development sporadically; its mutation rate accounts for around one percent and five percent in relation to familial Parkinsonism, respectively, making LRRK2 one of its more prominent genetic contributors to its association with Parkinsonism. LRRK2 plays an essential role in the GA [110,111,112,113]. LRRK2 mutants disrupt its integrity and vesicle trafficking. Furthermore, inactivation of LRRK2 leads to Golgi fragmentation and disrupted vesicle trafficking in hu man kidney proximal tubular epithelial cells via endocytosis and autophagy; additionally affecting endosomal system components including endocytosis and autophagy, with its dysfunction altering endosomal system components and organelles like endosomal and lysosomal function as well as synaptic vesicle trafficking, eventually altering neuronal function and synaptic plasticity (Figure 3).

Rabs, small GTPases that play an essential role in intracellular vesicular transport, serve as key switches that regulate membrane transport within eukaryotic cells. Rab proteins interact with their binding domain (RBD) to recruit Rab effectors into subcellular compartments via their binding domain, regulating vesicle formation, transport, and fusion processes [114,115,116]. Leucine-rich repeat kinase 2 (LRRK2) can directly phosphorylate Rabs; when overexpressed, 14 Rabs become substrates of endogenous LRRK2. Rab8, Rab10, and Rab29 (commonly referred to as Rab7L1) interact with LRRK2. Rab8a, Rab8b, and Rab10 function downstream while Rab29 is upstream; LRRK2 phosphorylates Rab29 in turn; these two interact in an intricate regulatory relationship; when overexpressed, Rab29 recruits mutant LRRK2 proteins into trans-Golgi network (TGN)-derived vesicles while simultaneously controlling its activation under lysosomal stress conditions. LRRK2 phosphorylates and recruits Rab8 and Rab10 proteins, with its activity and localization having an influence over Rab10 phosphorylation (Figure 3). Rab8 and Rab10, when phosphorylated by LRRK2, tend to accumulate around the centrosome, leading to insufficient centrosome cohesion [117,118,119]. Common LRRK2 mutations associated with Parkinson’s disease include G2019S and I2020T mutations as well as those in the ROC-COR domains. A number of familial Parkinson’s mutations increase autophosphorylation of LRRK2, possibly an indicator of its kinase activity. Rab GTPases such as Rab10 and Rab12 have been identified as key LRRK2 substrates within cells, with their phosphorylation intensifying when pathogenic LRRK2 mutants are introduced into healthy ones. The 14-3-3 proteins play an integral part in PD regulation [120]. Mutations affecting LRRK2 that disarm its interaction with 14-3-3 may increase levels of LRRK2, leading to accumulation. Rab32, Rab38, and Rab29 interact directly with LRRK2 to regulate its subcellular localization. Rab32 interacts directly with Sorting Nexin 6 (SNX6), a subunit of the retromer. Together with Rab38/38 and the retromer complex, this interaction regulates signaling pathways associated with LRRK2 activation; furthermore, a missense mutation of Rab32 has been linked with Parkinson’s disease (PD).

Furthermore Rab32, in tandem with Sorting Nexin 6/Retromer regulates localization of mannose-6-phosphate receptor on TGN which promotes Golgi trafficking [121,122].

## 7. Associations between Golgi Stress and ER Stress

The Golgi stress response occurs when the Golgi apparatus becomes overwhelmed. While not as extensively studied as endoplasmic reticulum (ER) stress, there is increasing evidence that various signaling pathways play a part in this reaction. TFE3 transcription factor binding is one such pathway, targeting transport proteins and structural proteins of the Golgi complex to alter the structure and function of the Golgi complex [123].

Under normal conditions, TFE3 is phosphorylated and located in the cytoplasm. Under Golgi stress conditions, however, TFE3 becomes dephosphorylated and translocates to the nucleus, where it activates Golgi-related genes, for instance when infections with dengue and Zika viruses trigger TFE3 translocation for activating the Golgi stress response [124].

Golgi stress responses involve various signaling pathways such as proteoglycans, mucins, mitogen-activated protein kinases/erythroblast transformation specific (MAPK/ETS), protein kinase R (PKR)-like endoplasmic reticulum kinase (PERK) and HSP47 pathways; they can lead to either apoptosis or provide protection from certain conditions such as cysteine deficiency in Huntington’s models [125]. Though the Golgi stress response has been linked with disease progression, its therapeutic value remains uncertain. Some studies indicate that inducing it could have beneficial results for human diseases, for instance, restoring redox balance in Huntington’s disease models or decreasing lung damage during acute lung injury simulations.

Fragmentation is an early step in neurodegenerative diseases and contributes to Golgi stress responses. Recent research indicates that mutations of the Trip11 gene, which encodes GMAP210, may cause both ER and Golgi stress responses in osteogenic cells; more research needs to be conducted into how Golgi and ER stress interact in disease pathogenesis. Glutathione (GSH) probes have been developed as a novel technique to measure Golgi stress, using fluorescent probes that detect and bind GSH as indicators. This novel method could offer additional insights into the link between Golgi and ER stress levels [126,127].

The Golgi apparatus located in cells can help withstand oxidative stress from either internal or external sources within their environment, including the direct production of reactive oxygen species by certain components of the stroma, while hypoxia can generate reactive oxidant species through dysfunction in mitochondrial electron transport and NADPH oxidase activity. Recent research using hydrogen peroxide to induce ROS-related oxidative stress revealed that trans-Golgi membrane tethers were especially vulnerable. When cells were treated with hydrogen peroxide, Arl1 degradation and detachment of the GRIP domain-containing proteins Golgin-97 and Golgin-245 from the trans-Golgi were observed. Hypoxia has also been demonstrated to influence the regulation of enzymes involved in nucleotide sugar synthesis and Golgi-localized glycosyltransferases, fucosyltransferases, and sialyltransferases. Furthermore, activating HIF-1a/HO-1 pathways through oxidative stress may increase expression of Golgi proteins such as GM130, Golgin-97, and mannosidase II [128,129,130,131].

## 8. Exploring Therapeutic Avenues for Golgi-Associated Disorders

While the exact processes and mechanisms associated with Golgi-related pathogenesis remain to be fully comprehended, current insights point out three specific research areas that could play a crucial role in targeting therapeutic targets for human diseases as well as understanding the mechanisms driving Golgi-related pathogenesis.

One area of research with great promise is identifying molecules that regulate Golgi structural changes, with many studies having identified candidate molecules such as GM130 and GRASP55 as possible candidates for further examination [132]. Fragmented Golgi structures are characteristic of many neurodegenerative and infectious diseases, but it is notable that ZEB1 and PAQR11, two EMT-activating transcription factors, encourage Golgi compaction instead of fragmentation in LUAD cells [133,134]. Golgi structures might also play an integral part in altered glucose metabolism. When intracellular ATP levels decrease following prolonged treatment with nonmetabolizable glucose analogs such as 2-deoxy-d-glucose, for example, Golgi bodies become smaller and denser, potentially altering metabolic activity in ways not previously predicted. Recent findings indicate that gastric cancer cells with more compact Golgi bodies exhibit greater metastatic capabilities. To harness Golgi-related changes with therapeutic potential for various human diseases, it is crucial to identify molecular switches that affect its steady-state structural features (i.e., fragmentation or compaction) and related processes of regulation [135].

Evidence continues to emerge of the importance of Golgi stress pathways in diseases associated with the Golgi. Stressful conditions such as energy shortages, pro-apoptotic effects, and DNA damage compromise Golgi structure and function through the cleavage of its structural proteins. As it has long been demonstrated, during apoptosis several proteins such as Golgin-160, GRASP65, p115, GM130, syntaxin 5, and giantin are cleaved by caspases to initiate Golgi fragmentation [136]. However, other stress response pathways and regulators still remain poorly understood. Lately, inside mammalian cells, six Golgi stress response pathways have been identified, such as the TFE3 pathway [137]. Unfortunately, specific sensors for each of these pathways remain unknown responding to Golgi stress could control degradation pathways including EGAD, GARD, and GOMED. Future studies that aim to develop strategies for targeted modulation of Golgi stress that is conducive to therapeutic applications should endeavor to: 1. Identify molecular sensors in each Golgi stress pathway; 2. Examine how these pathways identify their degradative targets.

Thirdly, GRASP55 has become an intriguing target in conditions associated with the Golgi. There are two vertebrate isoforms of GRASPs: GRASP55 and GRASP65. Studies have elucidated GRASP55’s unique role in various pathophysiological conditions despite initially being identified as a Golgi-resident protein [138]. GRASP55 depletion reduces a5b1 integrin protein expression in cervical and breast cancer cells; overexpression has also been found in invasive adenocarcinomas. Deficits in GRASP55 have been linked to decreased lipid droplet mobilization and, thus, defects in the absorption of lipid droplets, leading to defects in absorption. More recent findings showed that the GRASP55-dependent secretome is essential for the metastasis and angiogenesis of TP53-deficient LUAD metastasis as well as angiogenesis within tumor microenvironments [139]. GRASP55 responds to stress by altering its cellular location; for example, during times of ER stress, it is relocating from the Golgi to the ER, thus activating its UPS of CFTR. GRASP55 is released from the Golgi in response to glucose deprivation and acts as a bridge between autophagosome and lysosome fusion, but its exact role in diseases associated with the Golgi remains largely unknown [140].

Retargeting GRASP55 requires specific post-translational modifications to its C-terminal domain. Relocating it from ER and autophagosome/lysosome locations necessitates phosphorylating the serine 441 residue and de-O-GlcNAcylation of five serine/threonine residues, respectively; for this reason, developing treatments for diseases associated with GRASP55, such as cystic fibrosis, lipid metabolic diseases, or cancers, would benefit from small-molecule compounds capable of inducing either phosphorylation or de-O-GlcNAcylation, thus activating GRASP55, which can be beneficial [141].

Problems in membrane trafficking mediated by the Golgi are a hallmark of neurodegenerative disorders. An in vitro model that simulates early-stage epilepsy illustrates this fact; for instance, collapse of an ER-lysosome-Golgi network results in decreased processing of reelin protein, which contributes to seizures and seizure initiation/propagation; therefore, restoring this network could provide an innovative therapeutic strategy for alleviating epilepsy [142].

Animal models of Parkinson’s disease demonstrate how alpha-synuclein’s blocking of ER-to-Golgi membrane trafficking contributes to dopaminergic neurons’ degeneration and loss. Furthermore, increased expression of Rab1, which facilitates this pathway, may protect these neurons against alpha-synuclein-induced degeneration while improving survival and motor function control for enhanced therapeutic benefit.

ADAM10, a major alpha-secretase involved in amyloid precursor protein sheddling, limits beta-amyloid formation by controlling trafficking from dendritic Golgi outposts to synaptic membranes and is thus an integral factor of ADAM10 activity; increasing its activity via synaptic trafficking regulation could provide an effective treatment strategy against Alzheimer’s disease [143,144,145].

## 9. Conclusions

At its heart, understanding the Golgi apparatus—from its initial discovery in the 19th century until now being recognized as a central element in cellular biology—has been an exciting journey of scientific exploration. This remarkable organelle, with its distinctive organization and structure, has revealed multiple roles it plays, such as protein processing/sorting as well as lipid biogenesis, that are crucial in maintaining homeostasis within cells.

The Golgi apparatus’s involvement in neurodegenerative diseases, particularly Parkinson’s, has shed light on the intricate interplay between cell structures and disease pathology. Dysfunctional Golgi dynamics or modifications have been linked to disease onset or progression, underscoring its critical function for neuronal wellbeing.

Golgi stress and endoplasmic reticulum (ER) stress interact to provide new insights into our understanding of cell stress response mechanisms, giving us deeper insight into their roles as they relate to protein misfolding and accumulation, hallmark characteristics of many neurodegenerative diseases.

The Golgi apparatus is not just another cellular organelle; it is an active entity closely tied to multiple processes and disease states, making its role essential to health and disease. By uncovering its mysteries, we take one step closer to developing therapeutic approaches targeted at its dysfunction. Exploration of its historical background as well as its role in health and disease has underlined its importance in research, with its development being evidence of scientific understanding while at the same time promising innovative strategies against neurodegenerative disorders.

## Figures and Tables

**Figure 1 cells-12-01972-f001:**
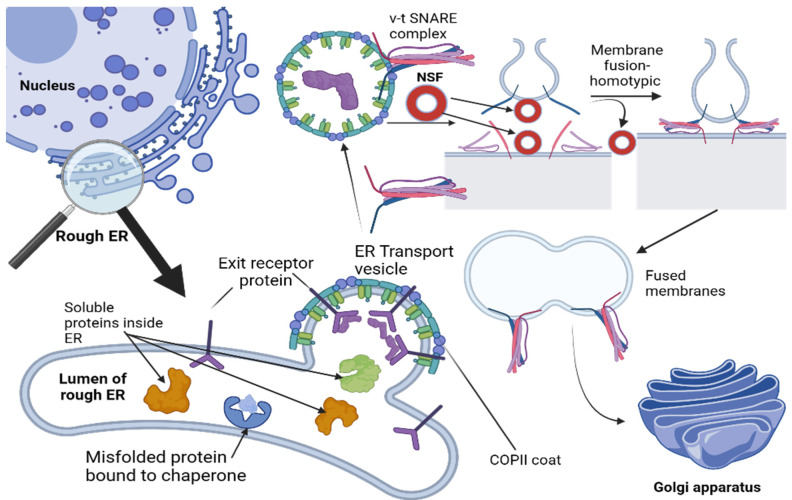
Membranes and cargo proteins leave the endoplasmic reticulum (ER), leaving behind transport vesicles packed with proteins by attaching themselves to the COPII coat. This binding involves exit signals on membrane proteins being trapped by the COPII coat and binding with it via their cytosolic tails; some membrane proteins trapped by it then act as cargo receptors to help package other proteins into transport vesicles for transportation. Membrane fusion and the formation of a continuous compartment involve interactions among matching v-SNAREs and t-SNAREs on adjacent identical membranes, beginning with the NSF separation of identical pairs on both membranes displaying matching SNAREs. Thereafter, homotypic fusion expands further by joining with similar-typed vesicles that display matching SNAREs. Once transport vesicles have left an ER exit site and shed their outer layers, they begin merging together. This process, known as homotypic fusion, involves membranes from within one compartment joining forces as opposed to heterotypic fusion, where separate compartments combine; it differs from heterotypic fusion, in which different compartments come together through merging. Like heterotypic fusion, homotypic fusion also requires two corresponding SNAREs, but unlike heterotypic fusion, both membranes contribute v-SNAREs and t-SNAREs, thus creating an even greater level of interaction than heterotypic fusion does.

**Figure 2 cells-12-01972-f002:**
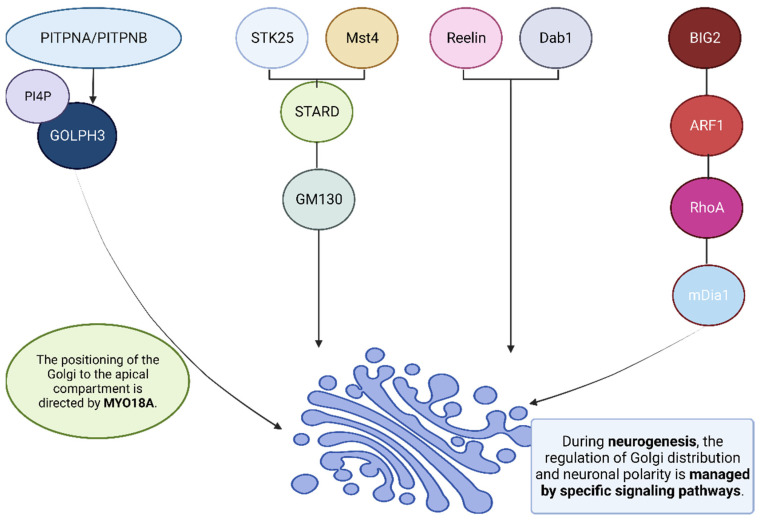
Signaling pathways play an essential role in organizing the Golgi apparatus and maintaining neuronal polarity during neurogenesis. First, lipid transfer proteins (PITPNA/PITPNB) augment GOLPH3 recruitment to the Golgi apparatus via PI4P signaling; which promotes its positioning towards MYO18A’s target compartment. Secondly, STK25 and Mst4, two downstream effectors of LKB1, possess the capacity to co-immunoprecipitate with STRAD and bind to Golgi matrix protein GM130, acting as key organizers within its organelles. Reelin and Dab1 also regulate the expansion of Golgi bodies into pyramidal neurons’ apical processes. Finally, the BIG2-ARF1-RhoA-mDia1 signaling pathway plays an integral role in controlling dendritic Golgi deployment and growth in newly created hippocampal neurons in adults.

**Figure 3 cells-12-01972-f003:**
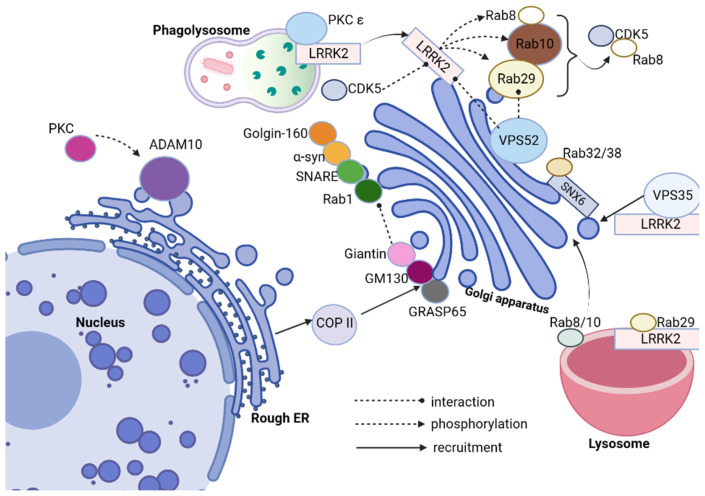
Leucine-rich repeat kinase 2 (LRRK2) acts as a central hub that interacts with numerous molecules, such as Rab family proteins, vacuolar protein sorting protein (VPS), Golgi outposts and Golgi posts (GOPs), cyclin-dependent kinase 5 (CDK5), Ppotein kinase C (PKC), and synapse-associated protein 97 (SAP97), among others. These interactions impact trans-Golgi network membrane transport and Golgi morphology which ultimately contribute to the development of pathological features associated with Parkinson’s disease (PD).

## Data Availability

All data is available online on libraries such as PubMed.

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
