# Peer review of "The Golgi Apparatus: A Voyage through Time, Structure, Function and Implication in Neurodegenerative Disorders"

_cells, 2023, doi:10.3390/cells12151972_

Round 1

Reviewer 1 Report

The manuscript by Mohan et al. reviews the Golgi Apparatus knowledge, from its discovery to the functions it perform in cells. Also, authors comment on the content of vesicles and their sorting into distinct compartments, and the consequences of fails in these processes for neurodegenerative diseases. The manuscript is well-written.

There are some concerns that can be addressed by the authors to increase the message of their manuscript; in general, it is difficult to know which reference is used to support the information provided. The figures are barely informative with respect to the title of the revision manuscript. Authors include information on yeast organization of Golgi and inheritance, and changes in parasites, but little information on the host cell during infections. This can be reorganized. The role of the Golgi in the neurological diseases is not clearly exposed in the manuscript. Authors could include a dedicated figure to this topic (which is in the title of the manuscript and therefore relevant for the take-home message).

Section 1:

Please, reference paragraph 1, lines 82-93

Please, reference paragraph 2, lines 94-103

Please, for sake of clarity, use only one name to designate same protein or complexes throughout the manuscript.

Section 2:

Please, reference paragraph 1, lines 132-140

Please, reference paragraph 2, lines 145-161

Please, reference paragraph 3, lines 165-179

Section 3:

Please, reference paragraph 3.1, lines 183-192

Please, reference paragraph 3.2, lines 214-216, 223-228, and 231-234

Please, reference paragraph 3.2, lines 251-255, 258-262 and 289-303

Please, reference paragraph 3.3, lines 308-316

Please, reorganize and refer the paragraph on COG complex: lines 383-396

Please refer the paragraph: lines 401-413

Please refer the paragraph: lines 416-440

Line 444: This is paragraph 4

Please refer the paragraphs: lines 469-472 and 473-483

Section 4 (Should be 5): paragraphs too long without inclusion of references after each sentence form lines 517-555

Sections 6 and 7 present a similar problem with references than the other sections.

Section 7, including the relationship with the ER should be upper in the text, unless authors correlate the stress process with any disease.

Figures

Figure 1: Please, identify each protein/lipid component in the schemes included in the figure. Mention the figure throughout the text.

Figure 2: It is not clear its relationship with a disease: please, include some biological context. Mention the figure throughout the text.

 Can authors discuss about the utility of assessing Golgi performance as a potential biomarker for diagnostic or prognostic of diseases?

The English language is OK. Some editing is required. 

Author Response

Thank you for reading our manuscript and for providing insightful feedback. I hope you'll find in the attached document that we adressed all of your comments.

Reviewer 2 Report

I enjoyed reading the manuscript. The report is well written and nicely summarizes Golgi Apparatus (GA) knowledge. I also appreciate the historical background of GA research results. However, I personally would shorten this chapter a bit to make the review more comprehensive.
Otherwise, the manuscript is suitable for publication.

The manuscript is well written.

Author Response

(The authors gave the same response as above.)

Reviewer 3 Report

This review by Mohan et al comprehensively covers historical and biological aspects of Golgi apparatus. It is written well overall, but readers will appreciate if there are more schematics that visually explain what is described in the review.

There are typos and grammatical errors throughout the text. And, there are several repetitive sentences (for example, line 473–475, 556–560, and 591-593). Also, I suppose it is wrong to use "Golgi cells", "Golgis", or "Golgi apparatus cells". Please carefully review and update the text.

Figure 1 is quite poorly drawn and explained: it is unclear which object represents which molecule or protein in the schematic. In addition, the figure legend describes Rab proteins, microtubules, and other Golgi components, but they should not be mentioned if not appearing in the schematic.

Figure 2 is better drawn compared to Figure 1, but it is quite unclear what each arrows mean. What do solid and dashed lines mean? Current schematic and figure legend is no different from just listing proteins, and there is not much good information presented in this figure. So, both figures need major update before publication.

There are typos and grammatical errors throughout the text. And, there are several repetitive sentences (for example, line 473–475, 556–560, and 591-593). Also, I suppose it is wrong to use "Golgi cells", "Golgis", or "Golgi apparatus cells". Please carefully review and update the text.

Author Response

(The authors gave the same response as above.)

Round 2

Reviewer 1 Report

The manuscript has very much improved. In particular, the revised figures and the new one. The document revised at this stage in "tracking changes" mode showed some possible typos to be edited (line of reticular "oreticular"; instructive "in structive").